# Reversal of Multidrug Resistance by Symmetrical Selenoesters in Colon Adenocarcinoma Cells

**DOI:** 10.3390/pharmaceutics15020610

**Published:** 2023-02-11

**Authors:** Bálint Rácz, Annamária Kincses, Krisztián Laczi, Gábor Rákhely, Enrique Domínguez-Álvarez, Gabriella Spengler

**Affiliations:** 1Department of Medical Microbiology, Albert Szent-Györgyi Health Center and Albert Szent-Györgyi Medical School, University of Szeged, Semmelweis utca 6, 6725 Szeged, Hungary; 2Department of Biotechnology, Faculty of Science and Informatics, University of Szeged, Közép fasor 52, 6726 Szeged, Hungary; 3Institute of Biophysics, Biological Research Centre, Eötvös Loránd Research Network, Temesvári krt. 62, 6726 Szeged, Hungary; 4Instituto de Química Orgánica General (IQOG), CSIC, Juan de la Cierva 3, 28006 Madrid, Spain

**Keywords:** multidrug resistance, P-glycoprotein or ABCB1, apoptosis, selenoesters, metastasis

## Abstract

Recently, selenium containing derivatives have attracted more attention in medicinal chemistry. In the present work, the anticancer activity of symmetrical selenoesters was investigated by studying the reversal of efflux pump-related and apoptosis resistance in sensitive and resistant human colon adenocarcinoma cells expressing the ABCB1 protein. The combined effect of the compounds with doxorubicin was demonstrated with a checkerboard assay. The ABCB1 inhibitory and the apoptosis-inducing effects of the derivatives were measured with flow cytometry. Whole transcriptome sequencing was carried out on Illumina platform upon the treatment of resistant cells with the most potent derivatives. One ketone and three methyl ester selenoesters showed synergistic or weak synergistic interaction with doxorubicin, respectively. Ketone selenoesters were the most potent ABCB1 inhibitors and apoptosis inducers. Nitrile selenoesters could induce moderate early and late apoptotic processes that could be explained by their ABCB1 modulating properties. The transcriptome analysis revealed that symmetrical selenoesters may influence the redox state of the cells and interfere with metastasis formation. It can be assumed that these symmetrical selenocompounds possess toxic, DNA-damaging effects due to the presence of two selenium atoms in the molecule, which may be augmented by the presence of symmetrical groups.

## 1. Introduction

The outcome of cancer treatment has improved significantly due to the recent advances in targeted therapy and immunotherapy. However, traditional combination chemotherapy combined with surgery or irradiation is still the most used method in treating cancer. Current chemotherapeutical agents have many drawbacks, including lack of specific toxicity, reduced delivery and effectivity in the hypoxic tumor environment, and the development of multidrug resistance (MDR) and stem cell-like phenotype [1].

Reactive oxygen species (ROS) are highly reactive molecules formed by the partial reduction of oxygen. Free radicals are continuously produced endogenously during cellular metabolism. There are also several exogenous stressors (e.g., radiation, xenobiotics, smoking) that can increase cellular ROS. Under physiologic circumstances, ROS are involved in signal transduction, inflammation, and immunity. However, in excess ROS may cause diseases; therefore, to balance the redox homeostasis cells use antioxidant defense systems [2,3,4]. 

The increased metabolic activity, mitochondrial dysfunction, and oncogene signaling, along with interacting with infiltrating immune cells, result in higher ROS levels in cancerous cells. Cancer cells may benefit from elevated ROS levels, as it can aid tumor progression and recurrence. Furthermore, to balance redox homeostasis, cancerous cells, especially cancerous stem cells, express high levels of detoxification enzymes and antioxidant enzymes, such as glutathione peroxidase and thioredoxin reductase. These defense systems may help cancer cells in the evasion of cell death and may lead to drug resistance. Still, the redox balance of cancer cells is fragile, exposing a potential therapeutic target in combating cancer [5,6]. 

Multidrug resistance poses a serious threat to antineoplastic therapy. MDR can be inherent or acquired and multiple mechanisms contribute to the development of the MDR phenotype. Intracellular mechanisms include increased DNA repair, evading apoptosis, altered drug uptake and metabolism, and the expression of efflux pumps and detoxification enzymes. Extracellular factors in the microenvironment, e.g., tumor vasculature, low pH, and hypoxia, are also key promoters of MDR phenotype [7]. According to the cancer stem cell concept, there is a small population of cancer cells with self-renewal and differentiating capabilities in the cancerous tissue that maintains the tumor. These stem cell like cancerous cells have slow cell-cycle kinetics and show increased expression of ATP-binding cassette (ABC) transporters and other genes related to MDR, thus being highly resistant to conventional chemotherapy [8]. ABC transporters are structurally related membrane transporters sharing similar intracellular nucleotide binding domains. Usually, they are present at various barrier surfaces (e.g., blood-brain barrier, hepatobiliary tract, gastrointestinal tract, renal tubules) and in the bone marrow to protect the cells against potentially harmful molecules by extruding them from the cytoplasm. In resistant cancers, the overexpression of ABC transporters, such as ABCB1 (P-gp or MDR1), presents a significant obstacle to chemotherapy, since many chemotherapeutic agents are substrates of these transporters [9]. Currently, the aim is to explore efflux pump inhibitors that act synergistically with chemotherapeutic agents to increase the sensitivity of resistant tumors to chemotherapy. Among non-competitive and competitive inhibitors of efflux pumps, compounds that modulate cellular ROS levels are attractive targets since ROS can regulate the expression of efflux pumps and can reduce the amount of ATP available for transport [10]. 

Numerous studies of the potential pharmacological applications and essential biological functions of selenium (Se) compounds have emerged in the past decades. Se as a component of selenoproteins is vital in the normal thyroid, brain, immune, and reproductive health. Se derivatives exert their bioactive effect through modulating the redox homeostasis of the cells. The activity of Se compounds is influenced by the concentration, chemical form (e.g., organic or inorganic, additional functional groups), and the metabolic activity of the organism. In lower concentrations, antioxidant and cytoprotective effects are prevailing; however, in higher doses, the prooxidant and cytotoxic effects become dominant [11,12,13,14]. The naturally occurring selenocompounds, such as selenate, selenite, selenomethionine, and selenocysteine, may be valuable in the prevention of cancer, as well as in sensitizing cancer to chemotherapy [15,16,17]. There are many forms of synthetic Se derivatives with promising anticancer activity or with MDR-reversing activity, including selenides and diselenides, selenocarbonyl compounds, and heterocycles containing selenium [18,19,20,21,22,23,24]. Among selenocarbonyl compounds, selenoesters have proved to be effective efflux pump inhibitors and chemosensitizer agents in various in vitro studies [25]. The underlying molecular mechanisms of selenoesters in MDR cancer cells are not fully known yet, but the available lines of evidence suggest that it may be related to the ability of the selenoesters to be hydrolyzed, releasing reactive selenium species (RSeS) that can be involved in intracellular redox reactions with thiols, free radicals, and reactive oxygen species [20,26]. The generation of RSeS could push this oxidative stress level in cancer cells above the critical redox threshold, leading to apoptotic or necrotic cell death [27]. This could explain the selective toxicity towards cancer cells compared to normal cells.

The aim of our study was to investigate the anticancer effect of symmetrical selenoesters on sensitive and resistant human colon adenocarcinoma cells expressing ABCB1 protein. The interaction of these selenium-containing compounds with doxorubicin was assessed with a checkerboard combination assay. The ABCB1 pump inhibitory and apoptosis-inducing effects of the derivatives were measured with flow cytometry. In order to demonstrate which genes have a key role in the activity of the compounds, whole transcriptome sequencing was carried out on Illumina platform upon the treatment of resistant cells with the most potent derivatives. The development of symmetrical derivatives is based on the principle that symmetry may favor the cytotoxicity and proapoptotic activity of compounds in comparison with asymmetrical derivatives [28]. This cytotoxicity related effect was observed in our previous study regarding methylselenoesters and dimethylselenodiesters [29]. The present study aims to evaluate whether the increase of selenium atoms and the increase of symmetry results in the enhancement of the activity regarding the respective unsubstituted non-symmetrical derivatives with one selenoester moiety [29,30]. The enhancement would be aligned with the previous reports that indicate the possible influence of symmetry on the anticancer activity [28,29]. 

## 2. Materials and Methods

### 2.1. Compounds

A total of 9 symmetrical selenoesters have been evaluated in this study, out of them 6 containing two selenoesters in their structure, the remaining 3 compounds possessing three selenium atoms. The selenoesters contain a functional group (methyl ketone, methyl ester, or nitrile) separated from the selenium atom by a methylene bridge. All compounds have a benzene ring as core, which can be *p*-disubstituted, *m*-disubstituted, or trisubstituted with the selenoester moiety. The structure of the compounds is shown in Figure 1. 

They were obtained following a previously reported procedure [30]. Briefly, one equivalent of phthaloyl chloride, isophthaloyl chloride, or 1,3,5-benzenetricarbonyl trichloride was reacted with 2 or 3 equivalents (in function of the number of carbonyls), of sodium hydrogen selenide generated in situ by reaction in an aqueous media of the adequate amounts of metallic powder of grey selenium with sodium borohydride. The crude reaction was filtered to remove borate salts, and the filtrate was reacted with 2 or 3 equivalents of the appropriate alkyl halide (2-chloroacetone, methyl 2-chloroacetate or chloroacetonitrile) to render the desired compound, which is filtered, washed, and purified as previously reported [30]. The synthetic procedure is graphically shown in Figure 2 for the three compounds with a 1–4 disubstitution pattern, using phthaloyl chloride as initial reagent. The compounds with a 1,3-disubstituted core departed from isophthaloyl chloride, and the trisubstituted derivatives were synthesized starting from 1,3,5-benzenetricarbonyl trichloride. 

The chemical reagents, solvents and materials required for the synthesis of the selenocompounds were purchased from Sigma-Aldrich Merck S.L.U. Spain (Madrid, Spain). Acros Organics and Alfa Aesar (brands of Thermo Fisher Scientific, Geel, Belgium); Honeywell-Riedel de Haën (Seelze, Germany), and Scharlab S.L. Spain (Sentmenat, Barcelona, Spain).

All compounds had a suitable purity to be evaluated in biological assays. Their purity was assessed using the technique of the elemental analysis, performed in a LECO CHNS-932 microanalyser (LECO Europe B.V., Geleen, The Netherlands). To consider a compound pure, it needs to have a deviation equal to or lower than 0.40% for each analysed element (carbon, hydrogen, nitrogen). The structure of the compounds was proven by means of NMR (both proton and carbon), IR, and MS [30]. 

### 2.2. Cell Cultures

The human colon adenocarcinoma cell lines, the Colo 205 (ATCC-CCL-222) doxorubicin-sensitive and Colo 320/MDR-LRP (ATCC-CCL-220.1) resistant to doxorubicin expressing ABCB1, were purchased from LGC Promochem (Teddington, UK). The cells were cultured in RPMI-1640 medium supplemented with 10% heat-inactivated fetal bovine serum (FBS), 2 mM L-glutamine, 1 mM Na-pyruvate, 10 mM Hepes, nystatin, and a penicillin-streptomycin mixture in concentrations of 100 U/L and 10 mg/L, respectively. 

### 2.3. Rhodamine 123 Accumulation Assay

The assay was conducted as described previously [25]. The tested compounds were added at 2 or 20 μM concentrations to the cells, and the samples were incubated for 10 min at room temperature. Verapamil was applied as a positive control at 20 μM. DMSO at 2% *v*/*v* was used as solvent control. The fluorescence of the cell population was detected with a PartecCyFlow^®^ flow cytometer (Partec, Münster, Germany). The fluorescence activity ratio (FAR) was evaluated as the quotient between FL-1 of treated/untreated resistant Colo 320 cells over treated/untreated sensitive Colo 205 cells using the following equation:$$\mathrm{FAR}=\frac{{\mathrm{Colo}320}_{\mathrm{treated}}{\;/\;\mathrm{Colo}320}_{\mathrm{control}}}{{\mathrm{Colo}205}_{\mathrm{treated}}{\;/\;\mathrm{Colo}205}_{\mathrm{control}}}$$

### 2.4. Checkerboard Combination Assay

A checkerboard microplate method was applied to study the effect of drug interactions between the compounds and the chemotherapeutic drug doxorubicin as described previously [31]. The assay was carried out on Colo 320 colon adenocarcinoma cells. The final concentration of the compounds and doxorubicin used in the experiment was determined in accordance with their IC_50_ values. Combination index (CI) values at 50% of the growth inhibition dose (ED_50_) were calculated using CompuSyn software (ComboSyn, Inc., Paramus, NJ, USA) to plot four to five data points at each ratio of compound and doxorubicin. CI values were determined based on the Chou–Talalay method, where CI < 1, CI = 1, and CI > 1 indicate synergism, additive effect (or no interaction), and antagonism, respectively [32,33].

### 2.5. Apoptosis Assay

The assay was carried out using Annexin V-fluorescein isothiocyanate (FITC) Apoptosis Detection Kit (Calbiochem, Merck KGaA, Darmstadt, Germany) according to the manufacturer’s instructions. The cell number of the resistant Colo 320 cells was adjusted to approximately 1 × 10^6^ cells/mL. The cell suspension was distributed as 0.5 mL aliquots (5 × 10^5^ cells) into a 24-well microplate and the cells were treated with the compounds at a final concentration of 0.25 μM (Se-K1, Se-K2, Se-K3), 0.5 μM (Se-C1, Se-C2, Se-C3) and 1 μM (Se-E1, Se-E2, Se-E3). The apoptosis inducer 12*H*-benzo[α]phenothiazine (M627) was used as a positive control at a final concentration of 20 μM and the assay was performed as described formerly [25]. The cells were stained with FITC-labelled annexin V and propidium iodide (PI) and then analyzed by PartecCyFlow^®^ flow cytometer. Based on their various stainability the necrotic (annexin V−/PI+), viable (annexin V−/PI−), early apoptotic (annexin V+/PI−), and late apoptotic (annexin V+/PI+) cells could be discriminated.

### 2.6. Preparation and Treatment of Cells for RNA Extraction

For RNA extraction resistant Colo 320 cells were applied. The density of the cells was adjusted to 5 × 10^5^ cells in 1 mL of RPMI medium and the cells were transferred in 1 mL aliquots into a 24-well plate. After an overnight incubation, the cells were treated with different selenoesters using a non-toxic concentration based on their respective IC_50_. The samples were prepared in triplicates using an untreated control and selenoesters as follows: ketone selenoester Se-K1 and Se-K2 were applied at 0.25 µM; methyloxycarbonyl selenoester Se-E2 was applied at 1 µM; cyano-selenoesters Se-C2 and Se-C3 were applied at 0.5 µM.

After the incubation of 24 h in a humidified atmosphere (5% CO_2_, 95% air) at 37 °C, the cells were removed from the 24-well plate using a cell scraper. Then, RNA isolation was performed by DeltaBio 2000 Kft. (Szeged, Hungary).

### 2.7. RNA Isolation and cDNA Library Preparation and Whole Transcriptome Sequencing

The cDNA library preparation and sequencing were performed by DeltaBio 2000 Kft. (Szeged, Hungary). Total RNA from the cell cultures were isolated with RNeasy Mini Kit (Qiagen, Hilden, Germany). The cDNA library for sequencing was generated with NEBNext Poly(A) mRNA Magnetic Isolation Module and NEBNext Ultra II RNA Library Prep Kit for Illumina (New England Biolabs, Ipswich, MA, USA). Illumina TruSeq HT indices (Illumina, San Diego, CA, USA) were used for the indexing. The nucleic acid concentrations of the samples were measured with Qubit 3 (ThermoFisher Scientific, Waltham, MA, USA) using the Broad Range RNA Assay Kit (ThermoFisher Scientific) for RNA samples and the High Sensitivity dsDNA Kit (Thermo Fisher Scientific) for the cDNA library. The quality was determined with Bioanalyser 2100 (Agilent Technologies, Santa Clara, CA, USA) using the Bioanalyser RNA 6000 Nano Kit for RNA samples and Bioanalyser HS DNA Kit for the cDNA library. The libraries were pooled and mixed with 1% PhiX DNA and sequenced on Illumina NextSeq 550 sequencing platform with NextSeq 500/550 High Output Kit v2.5 in 75 cycles (Illumina). The raw reads were deposited in the European Nucleotide Archive under the project number PRJEB57555.

### 2.8. Analysis of Transcriptomic Data

The quality of the generated paired-end reads was checked with FastQC 0.11.9 and MultyQC 1.11. Trimmomatic 0.36 and Cutadapt 2.10 were used for quality and adapter trimming with the default settings [34,35,36]. Mapping of the reads on the reference genome (Ensembl: GRCh38.p13) was performed with STAR Aligner 2.7.9a [37]. Transcript levels were determined with FeatureCounts 2.0.1 [38]. Principal component analysis and differential expression were calculated in R 4.1.2 with the edgeR package [39,40,41].

## 3. Results

### 3.1. Combined Effect with Doxorubicin

It was shown previously that all selenoesters possess strong antiproliferative and cytotoxic effects on sensitive and resistant colon adenocarcinoma cells. However, the methyl ester and nitrile selenoesters had potent cytotoxic activity on the cancerous cell lines being tumor selective because the derivatives were non-toxic on the normal MRC-5 cells [30]. Checkerboard combination was used to monitor the interaction of compounds with the known anticancer drug doxorubicin on the doxorubicin-resistant Colo 320 human adenocarcinoma cell line. The evaluated compounds showed different interactions from synergism to strong antagonism. From the evaluated compounds only the ketone moiety containing Se-K2 selenoester derivative exhibited significant synergistic interaction, other selenoesters with ketone moieties and nitrile moieties displayed antagonistic interaction. In the case of the selenoesters with methyl ester moieties, weak synergistic interaction was observed (Table 1).

### 3.2. ABCB1 Inhibition

The ABCB1-modulating activity of the compounds was assessed by rhodamine 123 accumulation assay on the doxorubicin resistant, ABCB1 expressing Colo 320 human adenocarcinoma cell line (Table 2). Verapamil, a known efflux-pump inhibitor, served as a positive control (20 µM) in the assay. The ketone and nitrile selenoester derivatives were capable to inhibit the ABCB1 efflux pump at 2 µM concentration with higher FAR values (FAR: 10–11) than the positive control verapamil (FAR: 3.22). This means that their ABCB1 efflux pump inhibitory activity, expressed in FAR at a 2 µM concentration, was 3.26- to 3.42-fold higher than the FAR measured for verapamil at 20 µM. However, the selenoesters with ketone moieties at 20 µM showed cytotoxic effects, confirmed by the high SSC (side scatter count) values, and decreased FAR values (Table 2). The methyl ester moiety containing derivatives showed weaker efflux pump inhibition compared to the positive control and other selenoester derivatives, with a FAR quotient range from 0.19 to 0.48. Cyano-containing symmetrical selenoesters showed an ABCB1 efflux pump inhibition higher than verapamil (FAR quotients are 1.06-1.97-fold higher) but lower than the one exerted by ketone-containing selenoesters (Figure 1).

### 3.3. Apoptosis Assay

Many chemotherapeutic agents exert their effect through interfering with the cell cycle via various mechanisms (e.g., alkylating DNA strands, inhibiting DNA synthesis, or inhibiting the mitotic apparatus), ultimately causing apoptotic cell death. However, cancerous cells may become resistant to apoptosis leading to the emergence of the MDR phenotype. We evaluated the apoptosis inducing ability of the selenoesters on the doxorubicin resistant Colo 320 cell line with Annexin V-FITC/PI double staining (Figure 2, Table 3). Regarding early apoptosis, selenoester Se-K1 with ketone moieties in 1,4 positions induced early apoptosis in 21.7% of the cell population being the most potent apoptosis inducer among ketone selenoesters (Table 3). The other ketone selenoesters Se-K2 and Se-K3 were also able to trigger early apoptosis; furthermore, all derivatives provoked late apoptosis that was more pronounced compared to the positive control M627 (Table 3). This may be explained by the potent ABCB1 inhibiting activity of these compounds, as ABCB1 may have a role in the regulation of the apoptotic pathway [42,43]. The methyl ester derivatives Se-E1, Se-E2, Se-E3 were also potent early and late apoptosis inducers being Se-E1 the most effective one that could induce early apoptosis in 37%, and late apoptosis in 30.7% of the cell population. Interestingly, these derivatives were not ABCB1 inhibitors, for this reason, the apoptosis induction is due to other mechanisms. Nitrile selenoesters (Se-C1, Se-C2, Se-C3) were the less active out of the three chemical groups of compounds; however, their moderate early and late apoptosis induction could be explained by their ABCB1 modulating properties.

### 3.4. Transcriptome Analysis

The transcriptomic data (Figure 3) revealed increased expression of *MT2A* gene that codes for cysteine rich proteins acting as ROS-scavengers protecting the cell from oxidative stress [44]. MT2A is associated with ROS-scavenging systems and its overexpression confirmed the redox modulatory activity of selenium compounds. The expression of *MDR1* gene encoding P-gp (ABCB1) was not altered significantly indicating that the efflux pump inhibitory activity of the compounds is not the result of decreased *MDR1* gene expression, there might be other factors that influence P-gp activity. Collagen XIII is a transmembrane protein involved in cellular adhesion and migration coded by *COL13A1* gene [45]. Upregulation of *COL13A1* is closely related to metastasis formation in some cancers. In the samples treated with Se-E2, a significant decrease in *COL13A1* expression was observed. Several other genes that may have a role in metastasis formation were also downregulated, such as *TNC* and *TENM1*, indicating that selenocompounds may also interfere with metastatic pathways. In addition, a marked decrease in the expression of *SOX21-AS1* gene was visible after Se-E2 treatment. SOX21-AS1 is highly expressed in various human cancers and may promote epithelo-mesenchymal transition and drug resistance in cancer, thus interfering with SOX21-AS1 may lead to increased sensitivity to chemotherapeutics and decreased metastasis formation [46,47]. Other genes that are associated with poor prognosis and tumor progression, such as *ODC1*, *ERN1,* and *ETS2*, were also downregulated primarily in the samples treated with Se-E2, Se-K2 and Se-C3 [48,49,50]. Selenocompounds decreased the expression of *AXIN2* and *WIF1*, indicating that selenocompounds may also modulate Wnt/β-catenin pathway. Furthermore, the downregulation of *LAMA3* and *CEP83* genes was detected in the samples.

## 4. Discussion

The main drawback of conventional chemotherapeutics is the lack of selective action on cancerous cells leading to unwanted side effects that may affect the chemotherapy treatment through dose reduction or discontinuation of chemotherapy. Additionally, the development of the MDR phenotype may also hinder the effectiveness of chemotherapy. ABC transporters, such as ABCB1, are thought to promote MDR along with apoptosis evasion and are expressed by cancer stem cells that are responsible for cancer progression and recurrence [9]. Thus, targeting ABC transporters may be an attractive approach to increase the effectiveness of chemotherapy and prevent relapse. Cancer cells maintain a fragile balance of elevated ROS levels that promote tumor progression and drug resistance, therefore modulating ROS levels in cancerous cells may result in cytostatic, cytotoxic, and MDR reversal effects [51]. Even a slight increase in ROS levels in these cells caused by prooxidants could pass the critical redox threshold, triggering apoptotic events in cancer cells [27].

Various selenoesters have exhibited antimicrobial, anticancer, and efflux pump inhibitory activity in previous series of studies [31,52,53]. In the present study, we evaluated the anticancer and efflux pump modulating activity of selenoesters that contain different moieties in different symmetrical orientations. The derivatives containing nitrile moiety demonstrated significant efflux pump modulating activity, whereas the methyl ester derivatives were able to produce a slight increase in doxorubicin cytotoxicity towards the resistant cell line. 

The ketone selenoesters Se-K2 and Se-K3 were able to provoke early apoptosis due to ABCB1 inhibition, as the MDR efflux pump ABCB1 may have a role in the regulation of the apoptotic pathway [42,43]. The methyl ester derivatives Se-E1, Se-E2, and Se-E3 were also potent early and late apoptosis inducers being Se-E1 the most effective one. Interestingly, these derivatives were not P-gp inhibitors. For this reason, apoptosis induction is triggered through other mechanisms. In contrast, the nitrile selenoesters could induce moderate early and late apoptotic processes that could be explained by their P-gp modulating properties. 

It can be presumed that the symmetry enhances the activity: the symmetrical ketone selenodiesters and selenotriester showed an efflux pump inhibitory activity comparable with the most active ketone selenoester, the 2-oxopropyl p-chlorobenzoselenoate [25]. This comparable activity was achieved without having the chlorine atom that significantly enhanced the activity of this derivative in comparison with the remaining asymmetrical oxoalkyl benzoselenoates evaluated so far. It would be interesting to evaluate in the future symmetrical compounds that contain also a halogen atom, as chlorine- and fluorine-substituted selenoesters have shown very promising activities in previous works [25,28,31,52]. Interestingly, apoptosis induction was achieved herein in the presence of symmetrical ketone derivatives at a very low concentration. It is noteworthy that the induction of late apoptosis in the case of the symmetrical derivative Se-K1 in the 59.1% of gated cells was demonstrated at 0.25 µM. Formerly, the most potent derivative induced late apoptotic events in a comparable or lower magnitude at a concentration 8-fold higher (2 µM) [25,31]. This fact supports the impact of molecular symmetry on anticancer activity.

The transcriptome analysis revealed that the efflux pump inhibitory activity of symmetrical selenoesters is not a result of the decreased expression of the P-gp encoding *MDR1* gene. The increased expression of *MT2A* gene may confirm the redox modulatory activity of the symmetrical selenium compounds. Furthermore, the expression of additional genes was also altered suggesting that the investigated compounds may interfere with pathways that are important in metastasis formation and in promoting tumor progression.

## 5. Conclusions

The most active symmetrical selenoesters tested herein have shown a more potent efflux pump inhibitory activity than the reference, according to the results of rhodamine assays. Some of them were also able to trigger apoptotic events in a potency comparable to the reference compound and furthermore, they affect the expression of relevant genes involved in the metastatic processes. This observed biological activity of the evaluated compounds may be explained through the modulation of redox homeostasis by selenium atoms of the selenoester backbone that is enhanced by the symmetrical moieties. The selectivity of the methyl ester and nitrile compounds towards cancerous cells may be a result of increased ROS production that damages the tumor cells more due to their fragile redox homeostasis.

## 6. Patents

This work with more details regarding the pro-apoptotic and multidrug resistant reversing activity of the selenocompounds is covered by the patent EP3628659A1 (filed on 28 September 2018 by Enrique Domínguez-Álvarez, Gabriella Spengler, Claus Jacob, and Carmen Sanmartín).

## Data Availability

Our raw sequencing data was uploaded to the European Nucleotide Archive under the bioproject PRJEB57555.

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
