# Peer review of "Reversal of Multidrug Resistance by Symmetrical Selenoesters in Colon Adenocarcinoma Cells"

_pharmaceutics, 2023, doi:10.3390/pharmaceutics15020610_

Round 1

Reviewer 1 Report

The article “Reversal of multidrug resistance by symmetrical selenoesters in colon adenocarcinoma cells” stands out for its high scientific level. It is very interesting research, well organized and presented in an easy-to-follow form.

I suggest the authors to add the reaction scheme according to which the compounds that are the subject of research in this article were obtained (to Materials and Methods).
How was the chemical structure of the synthesized compounds proven? (NMR, IR…). Just a short discussion
I also think that the authors should present the compounds in a table, with their main properties (molecular formulas, molar masses, melting temperatures), so the compound can be identified more easily.
One more recommendation: uniform writing for milliliter as mL.

I agree with the publication of the article after the minor revision.

Reviewer 2 Report

The paper by Bálint Rácz et al. is an interesting investigation about the multidrug resistance reversal in colon adenocarcinoma cells by symmetrical selenoesters.

The topic of the paper is worthy of investigation and well fits with the scope of the journal. Moreover, it is of interest for scientists working in a broad research field. Thus, it should be eventually published after addressing the following issues as per article subsections:

Abstract: in its present form it is too literal. Authors are encouraged to modify the section by reducing the introduction-like paragraphs and inserting some key results.

Introduction: authors well introduced the topic, the scope and the employed approach. They should improve the presentation of the expected advantages of their systems with comparison with the literature data. This can enhance the interest of readers.

Results: no specific comments to this section

Discussion: the discussion of results is well performed. Authors should consider the possibility to compare their results with different literature data to strengthen the advantages of their approach.

Minor points: please consider the possibility to move Scheme 1 to the introduction

Reviewer 3 Report

Racz et al describes reversal of MDR by selenium and its d/dx in colon cells expressing ABCB1 protein

The methods section can be further summarized

The discussion section is currently lacking in both depth and length. The authors need to elaborate how to interpret their findings in the context of the research question they are looking to address

(i)             So what was the anticancer effect of selenium and its derivatives on sensitive and resistant human colon cells? How was resistance inferred

(ii)            What did the checkerboard combination assay says about the observed trend? This if in line with doxorubicin. However, in clinical setting colon cancer patients are often treated with FOLFOX or FOLFIRI regime

(iii)          Was the results observed from whole transcriptome in line with the findings? Figure 3 requires a better presentation and clarity, how does this show the role of TNC and TENM1

(iv)           How can further studies verify the synergistic interaction observed

(v)            How were this taken into account P-gp inhibitors and apoptosis inducers? Or ABCBC1 inhibition in the related pathways? Does this need to be verified with PK or pathway experimental studies? How does Se-K2/K3 then provoke the early apoptosis in context?

(vi)           Are there confirmatory studies on the toxic effect of selenium on the backbone which is reported to be enhanced by symmetrical groups

Round 2

Reviewer 2 Report

The modified version of the paper properly addresses the comments. This reviewer is reccomending publication in its current form.